# Recent Advances in Alginate-Based Hydrogels for Cell Transplantation Applications

**DOI:** 10.3390/pharmaceutics16040469

**Published:** 2024-03-27

**Authors:** Alireza Kavand, François Noverraz, Sandrine Gerber-Lemaire

**Affiliations:** Group for Functionalized Biomaterials, Institute of Chemical Sciences and Engineering, Ecole Polytechnique Fédérale de Lausanne, 1015 Lausanne, Switzerland; alireza.kavand@epfl.ch (A.K.); francois.noverraz@epfl.ch (F.N.)

**Keywords:** alginate, cell transplantation, hydrogels, cell encapsulation, microencapsulation, microdevices, injectable hydrogels, regenerative medicine, 3D printing

## Abstract

With its exceptional biocompatibility, alginate emerged as a highly promising biomaterial for a large range of applications in regenerative medicine. Whether in the form of microparticles, injectable hydrogels, rigid scaffolds, or bioinks, alginate provides a versatile platform for encapsulating cells and fostering an optimal environment to enhance cell viability. This review aims to highlight recent studies utilizing alginate in diverse formulations for cell transplantation, offering insights into its efficacy in treating various diseases and injuries within the field of regenerative medicine.

## 1. Introduction

Regenerative medicine employs cell transplantation to trigger the repair and regeneration of diseased, dysfunctional, or injured tissues, utilizing stem cells or their derivatives [1]. This approach becomes crucial as adult tissues generally lack the inherent ability to regenerate, except for specific organs including the gut, cornea, skin, and liver [2]. For instance, in the case of acute liver failure where liver transplantation faces limitations due to organ shortage and the necessity for lifelong immunosuppression, hepatocyte transplantation emerged as a promising alternative for treatment [3].

Cell transplantation can be achieved through three main modalities: direct injection, scaffold-free cell sheet technology and encapsulation. Direct injection refers to the introduction of cell suspension at the targeted tissue or organ [4]. This minimally invasive procedure is commonly the preferred method for cell transplantation in clinical practice [5,6,7,8]. However, this approach frequently results in reduced cell viability within hours or days after transplantation [9]. For instance, very high death rates, up to 99%, were observed for neonatal cardiomyocytes directly injected in the center of injured hearts, in rodent animal models [10]. The microinjection process itself can subject cells to mechanical stress, leading to irreversible damage such as cell membrane rupture, initiation of apoptotic processes, and hindered viability [11,12,13]. This impact is especially notable in delicate cell types, like mature neurons [14]. Innovative technologies such as hollow microneedle arrays have shown promise in improving the direct injection process, reducing cell damage during injection [15]. But this approach might have limitations in certain cell types and applications. Furthermore, transplanted cells might rapidly disperse and lose their regenerative potential due to an unfavorable microenvironment such as harsh hypoxic conditions resulting from limited access to vasculature [16,17]. Another issue to consider is the immune rejection of transplanted cells [17], which necessitates the long-term use of immunosuppressive drugs, potentially leading to various severe side effects [18].

Direct injection is better suited for neurogenic diseases of the brain or injuries resulting from contusions in the spinal cord or peripheral nerves [19]. Nonetheless, challenges like low engraftment rates, poor cell retention and ineffective integration within surrounding tissue persist, limiting the overall efficacy of this method [20,21].

Cell sheet technology, as another method for cell transplantation, involves the cultivation of cells into thin, contiguous sheets, achieved by promoting cell growth and adhesion on temperature-responsive culture surfaces [22,23,24]. In this approach, the process of harvesting cell sheets is performed without the use of proteolytic enzymes. This method preserves cell surface proteins and intercellular junctions, thereby maintaining the integrity of cell interactions and the overall structural integrity of the sheet [25,26]. This method finds applications across various tissues and organs such as the liver [27,28], cornea [29,30,31], spinal cord [32], nerve [33,34], kidney [35], brain [36], heart [37], skin [38], and bone [39,40]. It has shown superiority in cell survival over 12 months compared to cell injection methods, with fewer apoptotic cells observed in cell sheets than in cell suspensions, as reported by Takeuchi et al. [41].

Application of cell sheets to cell transplantation suffers from limitations associated with their suboptimal mechanical properties, primarily dictated by extracellular matrix (ECM) synthesis, which make their manipulation and handling challenging [42]. Spontaneous contraction and shrinkage of cell sheets, arising from changes in cytoskeleton morphology after detachment, further complicate the process [43]. Additionally, the inability to use cell sheets as scaffold-free constructions in specific areas of tissue engineering poses a restriction [44].

Another promising alternative strategy for cell transplantation involves the cell encapsulation approach. Encapsulation refers to the envelopment of cells within a semipermeable matrix based on biomaterials, facilitating the diffusion of nutrients, oxygen, and the efflux of therapeutic proteins. Simultaneously, it serves as an immunobarrier, preventing the diffuse of larger molecules like antibodies and T cells, which could induce graft failure [45] (Figure 1). The encapsulation approach promotes improved integration of cells with surrounding tissue, thereby supporting the sustained preservation of transplanted cells. In general, biomaterials play a crucial role in enhancing cell survival by offering substantial mechanical support both during cell encapsulation and after transplantation into degenerative host tissues [2]. Furthermore, biomaterials serve as versatile platforms for incorporating growth factors, cell adhesion peptide sequence and essential drugs, contributing to improved cell viability. Additionally, biomaterial scaffolds promote the development of vasculature around encapsulated cells, ultimately enhancing their survival in the host tissue.

In this review, we will describe diverse techniques for cell encapsulation, focusing on the utilization of alginate as an exceptional biomaterial for cell transplantation.

## 2. Alginate

Cell transplantation has made notable progress with applying the benefits of synthetic polymers, such as polyethylene glycol and polyvinyl alcohol, as matrices that provide favorable environments for cellular integration. An advantage of synthetic polymers lies in their tunable properties and straightforward synthesis, enabling the preparation of hydrogels, solid scaffolds, and microparticles for diverse cell transplantation applications. Moreover, they exhibit superior mechanical properties, enhancing their suitability for this purpose [46]. 

However, it is essential to acknowledge that the use of synthetic polymers in cell transplantation is not without challenges. Synthetic polymers can frequently trigger chronic inflammatory or immunological reactions after transplantation and may pose concerns related to toxicity [47].

Hence, a promising alternative for cell transplantation involves the utilization of biopolymers characterized by excellent biocompatibility, biodegradability, non-toxicity, and the capacity to adsorb bioactive molecules. Biopolymers can be categorized into three classes based on their chemical structure and source: polysaccharides, including alginate, hyaluronic acid, chitosan, and dextran; proteins, such as collagen, gelatin, and silk; and microbial polymers like poly(lactic acid) and poly-(γ-glutamic acid) [48].

Alginate is a naturally occurring anionic polymer derived from brown seaweed, and several of its salts are classified as generally regarded as safe ingredients for oral administration by the U.S. Food and Drug Administration (FDA). Due to its outstanding biocompatibility, gelation properties, cost-effectiveness, and low toxicity, alginate has undergone thorough examination in biomedical applications, particularly in the field of cell transplantation. Alginate is a copolymer consisting of (1,4)-linked β-D-mannuronic acid (M unit) and α-L-guluronic acid (G unit) organized in block and atactic regions. The introduction of divalent cations such as Mg^2+^, Zn^2+^, Ca^2+^, Sr^2+^, and Ba^2+^ induces the binding of G unit blocks on adjacent chains, leading to the rapid formation of an ionic crosslinked network. Various cations exhibit different affinities for alginate-free carboxylate (COO^−^) groups. For instance, Ba^2+^ and Sr^2+^ display a higher affinity for alginate than Ca^2+^ [49]. The rapid and mild gelation conditions provide a significant advantage in preparing various forms of alginate gels for cell transplantations. For example, alginate serves as an excellent component, either alone or in combination with other biopolymers, for creating bioinks used in 3D printed scaffolds. It is also utilized in the preparation of solid anisotropic 3D scaffolds, injectable hydrogels, fiber, and microcapsules.

The presence of carboxyl and hydroxy groups in the alginate chain backbone facilitates the modification and functionalization of the polymer for modulating physicochemical properties, such as bioactivity, mechanical properties, gelation properties and biodegradation rate [50]. 

Purely ionically crosslinked alginate hydrogel exhibits limitations in stability, strength, and toughness, thereby restricting its potential applications in physiological environment. To address these challenges, a practical approach involves the introduction of chemical groups on the alginate backbone to enable covalent crosslinking, providing resistance to a broader range of conditions. Covalent crosslinking can be achieved through various functionalities, undergoing a broad range of chemical reactions such as click reaction, Diels–Alder reaction, Schiff-base reaction, radical polymerization, and Michael addition [51]. For instance, the modification of alginate by introducing PEG with thiol and acrylate functional groups allows for covalent crosslinking via Michael additions reaction, in addition to the existing ionic crosslinking [52,53]. This dual crosslinking strategy significantly improves the overall stability and performance of the hydrogel.

Oxidation of alginate is used to regulate the degradation behavior of alginate [54]. Previous studies have reported that alginate hydrogels crosslinked with calcium exhibit uncontrollable degradation kinetics [55]. Through oxidation, aldehyde groups are formed on the polymer backbone, enabling reactions with amines or hydrazides for blending with other biopolymers, such as gelatin. The resulting alginate conjugates holds promise for various applications in regenerative medicine [56].

A significant challenge in cell transplantation is pericapsular fibrotic overgrowth (PFO), which can noticeably hinder the diffusion rate of cell products, gases, and nutrients within the matrix, ultimately compromising the viability and function of the implanted cells [57]. Studies have indicated that the introduction of selected functional groups on the alginate backbone such as triazole [57], sulfate [58], and zwitterion [59] moieties can effectively mitigate PFO. Furthermore, the modification of alginate with anti-inflammatory drugs like ketoprofen has been shown to efficiently reduce PFO [60].

The modification of alginate with cell-adhesive peptides, such as the arginine-glycine-aspartic acid (Arg-Gly-Asp, RGD) sequence, can enhance cell adhesion which results in increased cell engagement with the matrix [61]. Additionally, the modification of alginate with galactose, a specific ligand for the asialoglycoprotein receptor (ASGPR) on the surface of hepatocytes, offers another approach to enhance cell adhesion. For instance, a recent study demonstrated that galactosylated sodium alginate effectively promoted the adhesive growth of HepG2 cells within an optimal concentration range [62].

Another significant motivation for alginate modification is to regulate its degradation, a critical factor as, for a number of applications, degradation ideally needs to synchronize with the deposition of the new cell-derived ECM [50]. For instance, alginate modified with the proline-valine-glycine-leucine-isoleucine-glycine (PVG↓LIG) peptide sequence becomes responsive to cleavage by matrix metalloproteinases (MMP) secreted by cells [63]. This modification allows for a controlled degradation process that aligns with cellular activity.

Therefore, the extensive range of modifications examined highlights the adaptability and promising potential of alginate as an outstanding biomaterial for advancing cell encapsulation in the field of regenerative medicine (Figure 2).

## 3. Cell Encapsulation Techniques

There are three main categories of cell encapsulation techniques based on the scale of encapsulation approaches and the size of final product: macroencapsulation, microencapsulation, and nanoencapsulation.

### 3.1. Macroencapsulation

Macroencapsulation generally refers to devices with a scale exceeding 1000 µm in at least one dimension, commonly known as macroscopic devices. These devices are characterized by retrievable chamber-based systems designed for the encapsulation of cells, offering optimal mechanical properties and excellent biocompatibility. These platforms must fulfill the dual function of allowing the permeation of essential nutrients, i.e., oxygen and amino acids while simultaneously shielding the encapsulated cells from the immune system [64]. The primary advantages of macroscopic devices lies in their retrievability and the ability to replenish cells when needed [65,66]. Such possibilities are crucial in the event of transplant failure or medical complications [67]. Moreover, the ability to retrieve these devices is a significant consideration in the regulatory approval process [68]. Nevertheless, it is worth noting that certain macroscopic devices loaded with cells, such as microneedles, have been utilized as transdermal patches without the need for implantation in the treatment of type 1 diabetes [69]. In this review, we present several macroencapsulation technologies such as devices based on porous membranes, anisotropic 3D scaffolds, 3D printing devices, hydrogel fibers, and injectable hydrogels. Each of these methods will be explained in detail in the following sections. 

#### 3.1.1. Device Based on Porous Membranes

The first type of macrodevice for cell transplantation employs a thin sheet fabricated either through electrospun nanofibers or solvent casting to create porous membranes [70,71,72,73]. These membranes, when layered, can adopt various configurations, such as tubular or planar shapes. Subsequently, alginate pre-gel is introduced into the membranes from the pore side, offering a substantial surface area that enables strong adhesion of the alginate hydrogel layer to the membrane. The key advantage of this macrodevice is its semipermeable membranes, shielding transplanted therapeutic cells from immune rejection and from escape while facilitating essential mass transfer to support cell survival and function. Notably, these membranes are robust, user-friendly, and easily retrievable.

Wang et al. introduced an intraperitoneal tubular device named the Nanofiber Integrated Cell Encapsulation (NICE) device [72]. This system is designed for the long-term preservation of transplanted insulin-producing cells, encompassing syngeneic, allogeneic, or xenogeneic rodent islets. The device comprises a highly porous and durable nanofibrous skin created through electrospinning a silicone-polycarbonate-urethane material, enclosing an alginate hydrogel core (Figure 1a–c). This dual-layer structure provides additional immunological protection and effectively prevents cell penetration while allowing maximum mass transfer. Moreover, tubular geometries provide simplicity in implantation and retrieval processes. When implanted in immunocompetent mice, the device loaded with syngeneic or allogeneic islets restored normoglycemia for up to 200 days. In a more challenging xenotransplantation model (human SC b-cell implanted in healthy dogs), the device retrieved 2 weeks post-transplantation still contained some functional cells. 

In another study, An et al. engineered a cylindrical encapsulation device called the nanofiber-enabled encapsulation device (NEED), utilizing a Nylon nanofiber porous shell [71]. This shell was immersed in a solution of alginate and subsequently subjected to vacuum treatment and crosslinking in the presence of BaCl_2_ solution. The resulting NEED was loaded with rat pancreatic islets and inserted into the peritoneal cavity of immunocompetent diabetic mice. Blood glucose regulation was maintained up to 8 weeks, demonstrating the ability of the device to preserve islets functionality over extended duration.

Very recently, Wang et al. introduced a microdevice comprising an asymmetric membrane with a gradient pore size transitioning vertically from over 50 µm to the nanoscale, constructed from polyether sulfone (PES) [70]. These microchannels served as chambers, enabling high-density cell loading and uniform cell distribution within the scaffold. Upon permeation of alginate hydrogel into the channels and subsequent gelation, a sealing layer formed, effectively impeding the infiltration of host immune cells into the scaffold. The devices were tested for biocompatibility using INS-1 insulin-producing cells and bone marrow-derived mesenchymal stem cells (BMSCs). The hybrid thin-sheet encapsulation system, approximately 400 μm thick, remarkably preserved allogeneic cells for over 7 months post-intraperitoneal implantation in immune-competent mice. In contrast, cells directly injected were completely cleared after 2 weeks.

Liu et al. introduced a device featuring an electrospun nylon nanofiber membrane coated with zwitterionically modified alginate—named SHIELD [73]. The device is composed of an islet loaded alginate gel immobilized in between two nanofibrous tubes (Figure 1d–h). The outer nanofibrous tube is additionally coated with zwitterionic alginate hydrogel to reduce fibrosis around the implant. This structure significantly improves the response delay for insulin secretion upon glucose level fluctuation. Additionally, this device offers retrievability and the flexibility to scale up in both radial and longitudinal dimensions while maintaining optimal mass transfer. Notably, it demonstrated its ability to support prolonged cell engraftment, effectively correcting diabetes in C57BL6/J mice with transplanted rat islets for a period of 399 days and in SCID-beige mice with human SC-β cells for up to 238 days.

#### 3.1.2. Anisotropic 3D Scaffolds

Designing macrodevices for cell transplantation in specific applications requires the creation of structures that mimic the complexity of natural tissues. The objective is to establish a microenvironment that optimally supports the growth and integration of transplanted cells.

For instance, 3D scaffolds with anisotropic properties could be essential in replicating the native longitudinal orientation of spinal cord nerve fibers in the context of Spinal Cord Injury (SCI) [74]. These specific topologies play a critical role in ensuring the alignment between proximal axons and distal stumps during the regeneration process. This alignment is particularly important due to changes in the spinal cord microenvironment and challenges posed by glial scarring, which often result in disconnection between proximal and distal neurons [75]. 

A multitude of hydrogels based on alginate bridges with anisotropic capillary structures with diameters smaller than 100 μm have been reported. These hydrogels have proven effective in directing axonal growth [76,77,78,79,80,81,82,83]. 

For instance, Günther et al. prepared alginate-based anisotropic capillary hydrogels measuring 2 mm in length and featuring capillaries with a mean diameter between 41 μm and 64 μm [81]. These hydrogels were seeded with BMSCs expressing brain-derived neurotrophic factor (BDNF), a well-established neurotrophin known for its efficacy in promoting neural protection and axonal regeneration after SCI [84]. Subsequently, the hydrogels were surgically implanted into a C5 hemisection lesion within the rat spinal cord. Their investigation revealed a remarkable increase in axon density within hydrogels seeded with BMSCs expressing BDNF (BMSC–BDNF) compared to those with control cells which expressed green fluorescent protein (GFP). Notably, this study also demonstrated substantial improvements in functional outcomes and enhanced electrophysiological conductivity. 

Another therapeutic approach involves the direct injection of neurotrophic factors such as BDNF to serve as a supplementary growth stimulus, facilitating axonal regeneration at the lesion site. For instance, Liu et al. developed alginate capillary hydrogels featuring evenly distributed capillaries at a density of 61 ± 4 channels/mm^2^, each with a diameter of 64.4 ± 6.2 μm [79]. Subsequently, Schwann cells were seeded onto the hydrogel, and BDNF was injected to enhance its regenerative potential. The results indicated that the injection of BDNF into the caudal spinal parenchyma led to substantial regeneration, in contrast to transplants lacking BDNF. The architecture of alginate hydrogels remained stable up to eight weeks post-implantation into a cervical hemisection. Additionally, the presence of Schwann cells in the host parenchyma facilitated the extension of descending axons across the hydrogel into the distal spinal parenchyma.

Another study highlights the significance of scaffolds featuring capillary structures, particularly for axonal guidance at damaged sites to facilitate regeneration [83]. The density of axons within alginate-based hydrogels was observed to correlate with the diameter of the capillaries. Larger capillary diameters were associated with reduced longitudinally oriented axon outgrowth.

#### 3.1.3. Hydrogel Fibers

In recent years, hydrogel fibers have emerged as a promising strategy to tackle challenges in cell transplantation [85]. Scaling up the production of fibers with a uniform size-controlled distribution is feasible and compatible with current good manufacturing practices, and their geometry indeed provides advantageous properties for efficient mass transfer.

The alginate fiber stands out as one of the most renowned macroscopic devices extensively used for loading cells in various cell transplantation procedures [65,86,87,88,89,90,91,92,93] as well as a microbioreactor for cell culturing for various applications [94,95,96,97,98].

Takeuchi et al. conducted a study involving the fabrication of barium alginate hydrogel fibers encapsulating primary rat islets [99]. They investigated the impact of the hydrogel fiber diameter on both biocompatibility and in vivo functionality of the encapsulated islets. Their findings revealed that the xenogeneic islet-laden fiber-shaped hydrogel implanted to immunocompetent male C57BL/6NCrSlc mice, effectively reduced in vivo cellular deposition, particularly when the hydrogel diameter exceeded 1.0 mm. This approach was utilized for achieving long-term glycemic control in diabetic mice.

One approach to designing macrodevices for cell transplantation involves utilizing reinforced fibers exceeding 1 mm in size to ensure adequate mechanical strength, facilitating portability and retrieval of the device [65,90,100]. These fibers are subsequently coated with alginate, incorporating the cells of interest, followed by the crosslinking of the alginate layer.

An et al. developed a method in order to reinforce mechanical strength that involved the creation of nanoporous-coating nylon threads derived from poly(methyl methacrylate) [65] (Figure 2a–d). These threads are used to crosslink an alginate solution containing suspended islets and the resulting system was called TRAFFIC (thread-reinforced alginate fiber for islets encapsulation). In this study, the rat islet-loaded fiber gel (~1-inch length) was surgically implanted into the peritoneal cavity of diabetic C57BL/6J mice. Remarkably, just two days post-implantation, the mice exhibited a swift return of their blood glucose levels to the normal range, which they maintained for a duration of four weeks. Neat alginate fibers were prepared for comparison with TRAFFIC. The incorporation of a central thread in TRAFFIC demonstrated superior mechanical properties, making it easier to handle and retrieve in comparison to the plain alginate fibers. Additionally, to scale up the system, they implanted the extended alginate fibers (~10-inch length) between the liver and diaphragm of larger animals such as dogs through laparoscopic procedure. The islets extracted from these implants maintained their viability and continued to secrete insulin in response to fluctuations in glucose levels.

Ernst et al. developed a unique encapsulation structure for islet transplantation using a methodology similar to the aforementioned study [90]. Their design consists of elastomer-reinforced interconnected toroidal hydrogels. The device features elastic scaffolds shaped like interconnected donuts, coated with a layer of alginate hydrogel seeded with cells (Figure 2e–g). These toroidal hydrogels offer a larger surface area compared to traditional spherical capsules created from a similar volume of hydrogel. Glucose-stimulated insulin secretion (GSIS) analysis demonstrated effective diabetes correction for up to 12 weeks when these hydrogels were transplanted into the intraperitoneal cavity of C57BL/6 mice. Moreover, complete retrieval of the device was achieved, highlighting a proof-of-concept for a novel retrievable device in β cell replacement therapies.

In a recent study, researchers introduced an islet-encapsulation device featuring a twisted nylon surgical thread coated with an islet-seeded alginate hydrogel [100] (Figure 2h–j), following a similar approach to a previously reported method [65]. To facilitate the formation of new blood vessels (neovascularization) around the implantation site, they induced controlled inflammation by implanting a nylon catheter temporarily, 4–6 weeks before transplantation. After removing the catheter, the device was transplanted into the site. This prevascularized cavity provided an optimally oxygenated environment for the encapsulated islets, notably boosting their survival and functionality.

Recently, alginate microtubes were introduced to facilitate the culturing of human pluripotent stem cells (hPSCs) and their subsequent differentiation into neural stem cells (NSCs) [91,95,101]. For instance, Lin et al. recently demonstrated a scalable technique employing a microextruder to encapsulate hPSCs in alginate microtubes [91]. Over a 12-day study, the encapsulated hPSCs expanded and differentiated into NSCs with excellent viability (~95%), high purity, and yield. Their findings suggest that the expansion rate of NSCs in 3D microtube is approximately 250-fold higher compared to traditional 2D culture substrates. This approach holds significant promise for advancing cell transplantation methodologies, particularly in generating NSCs for therapeutic purposes.

In another study, Naficy et al. engineered a tubular hydrogel-based device using hydrophilic polyurethane and polypropylene glycol, featuring a tunable porous structure (~150–600 μm) [102]. This device was coated with an alginate solution and loaded with both human insulin-producing pancreatic β-cell lines and porcine neonatal islet cell clusters. The device loaded with the β-cell line demonstrated a faster release of insulin in HEPES buffer as opposed to RPMI 1640 medium when exposed to a 33 mmol·L^−1^ glucose solution. Moreover, islet cell clusters exhibited superior performance, by doubling the insulin release compared to β-cell lines over a 24 h period when exposed to a 33 mmol·L^−1^ glucose culture medium. In both cases, the encapsulated cells displayed robust viability and retained their functionality in the presence of high glucose levels.

Evron et al. introduced a retrievable macroencapsulation device that contains islets within an alginate slab [103]. This device features a replenishable gas chamber supplying exogenous oxygen to prevent damage to transplanted islets caused by the consumption of pO_2_ within the gas chamber. Implanted encapsulated islets, with a surface density reaching as high as 4800 IEQ/cm^2^, in diabetic rats, sustained normoglycemia for over 7 months. Additionally, these islets exhibited near-normal intravenous glucose tolerance tests.

In a different study, human-stem-cell-derived pancreatic beta cells (hSC-βs) were encapsulated in lotus-root-shaped devices (LENCON) based on alginate, featuring sizes of 1 and 6 mm diameters [104]. These devices were transplanted into the abdominal cavity of mice for a long-term evaluation (>1 year). The findings revealed that the smallest devices experienced increased cellular deposition, resulting in a thicker fibrotic layer on their surface. Remarkably, recipients of LENCON transplantation exhibited normoglycemia within 10 days post-transplantation. Furthermore, the glucose tolerance levels in mice that received LENCON transplants remained comparable to those of normal mice even after 100 days following transplantation.

#### 3.1.4. 3D Printing Devices

Recently, the growing demand for organ transplants has driven the exploration of additive manufacturing technologies, notably three-dimensional (3D) printing. As a result, numerous studies have adopted the 3D printing approach to create macro-3D objects for cell transplantation [105,106,107,108].

The use of 3D printing for creating macrodevices brings numerous advantages. With a precisely defined structure, there is an increased surface area that improves mass transfer. Furthermore, 3D printing enables precise control over porosity, potentially reducing issues related to cell crowding [109]. The 3D printed devices have the capacity to facilitate the formation of a vascular network around encapsulated cells, resolving the challenge of effectively supplying oxygen and nutrients to the trapped cells [110,111]. This is a significant parameter, since hydrogel microcapsules typically do not promote vascular network formation after transplantation [112]. 

In general, certain 3D objects are structured resembling reservoirs, frequently employing alginate as the primary material to support and encapsulate cells within these devices [113]. For instance, ongoing research endeavors focus on permeable membranes using various materials like polyamide, polylactic acid, and polyurethane. These membranes are engineered to provide mechanical protection, user-friendly handling, and effective retrieval of cells [114,115,116,117].

Espona-Noguera et al. developed a 3D printed semipermeable polyamide macroencapsulation device [114]. In this system, the alginate hydrogel serves as an immunoprotective supportive matrix wherein the β-cells are embedded. Their study investigated the impact of hydrophilicity and porosity surface devices, revealing that the hydrophobic surface with smaller pore sizes exhibited superior cell viability values.

Another study featured encapsulated cells in alginate-poly-L-lysine microcapsules (~400 μm in size) into 3D printed porous polyamide macrocapsules (>50 µm pore size). This combination demonstrated favorable outcomes, revealing good cell viability, intact DNA, and efficient production of therapeutic products [118] (Figure 3a,b).

In a recent study by Chen et al., a 3D-printed scaffold was designed using sodium alginate and gelatin methacrylate [119]. Their investigation into the hydrogel’s modulus revealed that lower alginate ratios in the hydrogel composition led to reduced deformation when subjected to compression forces, simulating the brain tissue environment. The degradation of these hydrogels in the presence of collagenase II was notably higher at 95.73% compared to 10.90% in PBS after 72 h. Furthermore, they assessed the impact of cerebrospinal fluid (CSF) flow on cell retention using CSF MRI data post-transplantation of NSCs within an in vitro model simulating CSF flow. NSC-loaded hydrogel scaffolds cultured for seven days in neuronal differentiation medium demonstrated enhanced cell retention following CSF flush. Their findings from the transplantation of NSC-loaded hydrogel scaffolds into male SD rats showed significant neuroprotective effects against TBI. Notably, this intervention led to reduced microglial activation and neuronal death during the acute phase, highlighting its potential for neuroprotection post-TBI.

An innovative 3D printed reservoir was engineered, comprising a ring-shaped polycaprolactone (PCL) scaffold with heparin covalently conjugated to its surface in order to facilitate the loading of vascular endothelial growth factor (VEGF) within the scaffold [120]. This structure enclosed an alginate hydrogel loaded with human islets in its core (Figure 3c–e). The PCL scaffold was constructed using fibers measuring approximately 192 ± 14 μm in diameter, maintaining a fiber spacing of approximately 380 ± 29 μm. Upon implantation into a chicken chorioallantoic membrane, the potential for vascular development was evidenced by the formation of new blood vessels in the surrounding tissue and even on the surface of the 3D reservoir. The islets embedded within the scaffold maintained their functionality, secreting insulin in response to glucose stimulation with absolute secretion values similar to control islets.

In 3D bioprinting, one approach to cell encapsulation involves blending the cells of interest with a biocompatible material known as bioink to create printed structures. These structures provide an ideal microenvironment for stem cells, facilitating transplantation, proliferation, and differentiation.

Natural polymers like alginate, chitosan, collagen, agarose, gelatin, silk, and decellularized extracellular matrix (dECM) have gained prominence in recent years [121]. They are often used independently or in combination with other substances as the primary constituent of bioinks for various bioprinting applications.

Pure alginate inks often suffer from poor printability and tend to diffuse after 3D printing through the nozzles, leading to structural deformation due to their low mechanical strength [122]. However, printing alginate components into CaCl_2_ solutions allowed to maintain the structural integrity of 3D scaffolds during the printing process [123].

Blending alginate with other polymers like methylcellulose and polyvinyl alcohol (PVA) significantly enhances the viscosity of the alginate solution. This enhancement enables precise deposition and facilitates straightforward crosslinking with CaCl_2_ after the printing process [124]. A microporous 3D printed hydrogel was produced from ionically crosslinked alginate and methylcellulose loaded with rat islets [125]. Methylcellulose was gradually released over time, leading to the creation of a microporous structure. Cell viability and biological function were assessed, revealing viability levels close to 80%, with sustained glucose responsiveness observed for a period of seven days within the culture medium.

**Figure 3 pharmaceutics-16-00469-f003:**
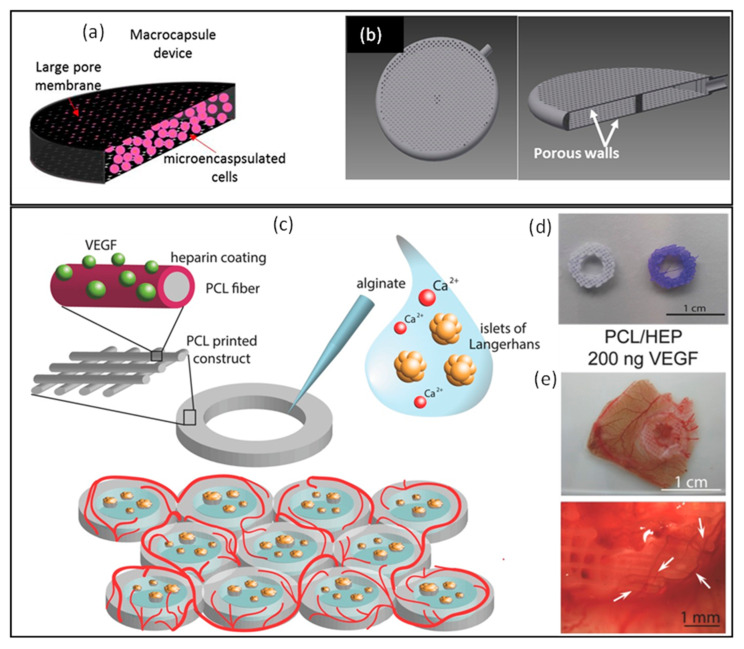
(**Upper panel**): Illustration outlining the double encapsulation approach within 3D macrocapsule devices (**a**), the image displaying the device, displaying both a hole and a cross-section of the device (**b**), adapted with permission from [118]. (**Lower panel**): A schematic representing the plotted scaffold involves a structure made of 3D-printed polycaprolactone rings with covalently coated heparin on their surfaces. The inner part of the structure encapsulates Islets of Langerhans using alginate hydrogel. Multiple constructs can be 3D printed adjacent to one another in a honeycomb arrangement, enhancing the available surface area for loading islets and promoting the scaffold’s revascularization (**c**). PCL and PCL/heparin plotted scaffolds (**d**), the promotion of neovascularization within heparin-coated scaffolds with a concentration of VEGF set at 200 ng, arrows indicate the blood vessel formation (**e**), adapted with permission from [120].

Another approach to prevent collapsing during printing involves the incorporation of nanofibers into the bioink formulated with alginate. One such example features the use of nanofibrillated cellulose within an alginate solution to create 3D bioprinted scaffolds known for their shape stability [126,127]. An additional example involves the incorporation of polylactic acid (PLA) nanofibers into a bioink formulation consisting of an alginate hydrogel and human adipose-derived stem cells (hASC) [128].

Moreover, the viscosity of an alginate-based bioink relies on variable parameters, including alginate concentration, its molecular weight, and the density and phenotype of the cells involved. It is crucial to consider these parameters to optimize cell viability in the printing process [129]. Alginate alone and in combination with various polymers has been extensively utilized as a bioink for cell transplantation [125,129,130,131,132,133,134,135,136,137,138,139,140]. For example, while gelatin methacryloyl (GelMA) upon light exposure forms chemically stable hydrogels for cell encapsulation, it was observed that creating uniform and stacked cell-laden fibers using low-concentration GelMA hydrogels can pose challenges for regular printing processes in certain cases [141]. Combination with physically crosslinked alginate fibers allowed the formation of complex scaffolds which supported the migration of human umbilical vein endothelial cells (HUVECs) and the synchronous beating of cardiomyocytes [133].

Bioprinted scaffolds based on alginate have demonstrated the potential to enhance vascular network formation, consequently improving cell survival [130,136,142]. For instance, Idaszek et al. developed two distinct bioinks by incorporating pancreatic dECM powder and fibrinogen into alginate to mimic pancreatic and vascular structures, respectively [136] (Figure 4g). These bioinks were loaded with porcine pancreatic islets and a combination of human mesenchymal stem cells (HMSCs) and HUVECs to simulate vessel formation. Glucose-stimulated insulin secretion revealed that the encapsulated cells supported the endocrine function of porcine pancreatic islets. Furthermore, in vitro investigations confirmed the formation of intercellular junctions and vessel-like structures by CD31-positive cells.

One advantage of utilizing bioink in bioprinting is the layer-by-layer deposition process, enabling the precise arrangement of various cell types and bioactive factors in a specific order [143]. This capability is instrumental in a wide range of applications in cell transplantation. For instance, multiple studies have highlighted the beneficial effects of including supporting cells like MSCs or endothelial cells in islet culture, demonstrating enhanced islet function and survival rates [144]. Liu et al. developed an innovative approach by fabricating core–shell 3D macroporous structures using an alginate-gelatin bioink to address key limitations in islet transplantation protocols [107]. Their strategy aimed at enhancing revascularization and providing immunoisolation for improved outcomes. The study involved co-encapsulating endothelial progenitor cells (EPCs) and T regulatory cells (Tregs) in the shell layer, while the core part of the printed 3D scaffold encapsulated islets through coaxial printing (Figure 4a–f). EPCs and Tregs were incorporated to potentially boost pancreatic islet engraftment by enhancing revascularization and suppressing the immune response. Another recent study highlights reduced apoptosis and increased expression of ECM molecules and VEGF in rat islets when bioprinted alongside endothelial cells, as evaluated through transcriptomic analysis [142].

Recently, Abadpour et al. presented novel findings on the supportive role of adipose-derived stem cells (ASCs) in pancreatic islets within 3D bioprinted scaffolds comprised of alginate and nanofibrillated cellulose [126] (Figure 4h). Their scaffold design features a distinctive double-layered structure, hosting mouse or human pancreatic islets in the top grid layer and human ASCs in the lower layer. The study illustrates that human ASCs notably enhance islet behavior and sustain their long-term function throughout a 60-day intraperitoneal transplantation in a diabetic mouse model. Upon scaffold recovery, the presence of ASCs within the scaffolds remained evident even 60 days after transplantation.

In Table 1, a summary of various types of devices for macroencapsulation has been presented.

#### 3.1.5. Injectable Hydrogels

Another approach to cell transplantation involves direct injection in the recipient body of a polymer solution containing the cells of interest, which undergoes subsequent rapid gelation at the injection site [152]. In this system, the polymer solution, acting as a viscous material, serves three crucial purposes: it protects the cells from the detrimental mechanical forces experienced during injection; it acts as a scaffold for local cell adhesion [153,154]; and it provides protection to transplanted cells against local inflammation and surrounding macrophages [155]. This method has several advantages, including minimal invasiveness of the implantation procedure and ease of administration [156].

In general, cell transplantation using this method involves two approaches. The first approach is based on the introduction of active residues or catalysts in the precursor polymer solution, which will initiate in situ chemical reactions for the formation of covalently crosslinked hydrogels [157,158]. However, this method is not adapted to the transplantation of proliferating cells as the hydrogel scaffold will not degrade over time [159,160]. To address this concern, the implementation of degradable crosslinking points, such as degradable peptides, offers the possibility to modulate the durability of the resulting hydrogel in vivo [161,162,163]. For example, Phelps et al. introduced a bioactive hydrogel designed specifically for islet transplantation, composed of a maleimide-functionalized 4-arm PEG macromer (PEG-MAL), crosslinked with a cysteine-flanked protease-degradable peptide known as GCRDVPMSMRGGDRCG (VPM) [161].

The second approach involves injectable hydrogels formed through physicochemical interactions, which do not require crosslinking agents for in situ reactions, for example, hydrophobic bonding interaction, host–guest interaction, self-assembly interaction and electrostatic gelation/ion-induced gelation [158]. Because these hydrogels degrade rapidly, they do not compromise the survival, proliferation, and biological functions of transplanted proliferating cells in vivo [164].

Alginate has been widely utilized in the production of injectable hydrogels for various biomedical applications [159,165,166], including cell transplantation [164,167,168,169]. Its biocompatibility and ability to form gels under mild conditions such as simple physicochemical interactions with other biopolymers (e.g., gelatin [164,170,171,172], collagen [173,174,175,176]), promote the delivery of encapsulated cells to the target site within the host body. [164]. The shear-thinning nature of alginate further aids in reducing shear stress during injection by creating a lubricating layer on syringe walls, resulting in more uniform flow velocities across the syringe [177,178]. The presence of alginate gel during in situ cell encapsulation has shown promising outcomes in enhancing cell retention. For instance, alginate increased cell retention up to 60% after injection into a dynamic environment like rat heart, while saline controls exhibited only 9% retention after 24 h [179]. Similarly, murine bone derived MC3T3-E1 cells encapsulated in gelatin/alginate/ferric ions injectable hydrogels showed high retention rates (82.5% after one day) after injection in the back subcutis of mice, as compared to cell suspensions in PBS (3.27% retention after one day) [164].

Injectable hydrogels based on alginate have been extensively studied in cell transplantation protocols for the treatment of various diseases, with a particular focus on SCI, myocardial infarction (MI) and traumatic brain injury (TBI). In the following section, we aim to highlight the most relevant studies in this field.

#### Nervous System Stem Cell Transplantation Therapies

A promising approach to the treatment of SCI involves the use of stem cell-derived therapies, which aim to either replace lost neuron populations or provide supportive glial cell types. Scientists have developed injectable hydrogel systems specifically tailored to enhance the survival, engraftment, and interaction of transplanted cells with the host tissue, aiming to improve the functional behavior of both neural and glial cells within damaged nervous tissue [180].

This type of injectable hydrogels represent a promising method for the treatment of severe SCI because they possess the capability to fill the injured defect and adapt to the irregular geometry of the contusive lesion with minimal surgical intervention [181]. Moreover, the direct injection of alginate hydrogels into the adult rat spinal cord has not demonstrated any incitement of an inflammatory response [76]. Matyash M. et al. demonstrated that alginate hydrogels serve as an adhesive matrix for neurons and promote axonal outgrowth [182]. Remarkably, these hydrogels support axonal growth without any modification or functionalization of extracellular matrix components, highlighting the potential influence of mechanical properties in controlling neurite outgrowth. Several comprehensive reviews reported applications of injectable hydrogels for SCI treatment [183,184,185].

Xing et al. developed an injectable alginate hydrogel loaded with the anti-inflammatory drug diacerein, and incorporating poly(styrene sulfonate) (PSS) and carbon nanotubes (CNTs), which showed an excellent conductivity and low cytotoxicity for SCI treatment [186]. Moreover, these hydrogels demonstrated significant potential in reducing astrocyte hyperactivity and mitigating inflammation.

An essential aspect of hydrogels is their ability to replicate the mechanical properties of native SP tissue. By carefully controlling the concentrations of alginate and Ca^2+^ ions, it is possible to produce alginate hydrogels that closely mimic the elastic modulus of native central nervous system (CNS) tissue [187]. McKay et al. reported that following the in situ formation of an alginate gel in the damaged area, the incorporation of chitosan and genipin may enhance the physicochemical properties of the hydrogel and modulate the behavior of astrocytes [188]. In addition, in vitro study demonstrated the remarkable potential of alginate/chitosan/genipin hydrogels, known for their high sensitivity to Ca^2+^ composites, in regulating astrocyte behavior. These hydrogels exhibited the capacity to reduce Ca^2+^-related secondary neuron damage, displaying a promising avenue for preventing such complications during acute SCI.

The issue of ischemia and hypoxia disrupting the balance of energy metabolism is a common concern in SCI cases. Alginate hydrogels were investigated for their potential to deliver regulating drugs such as fibroblast growth factor 21 (FGF21) at the site of injury. For example, in the context of repairing hemisection spinal cord injuries (HSCIs), Zhu et al. applied an alginate hydrogel containing dental pulp stem cells (DPSCs) as supplement exogenous neurons and FGF21 [189]. In areas of damage, the influx of calcium into nerve cells can disrupt regular cellular processes, potentially causing cell injury and ultimately leading to cell death. Interestingly, alginate showed the capacity to prevent calcium overload by absorbing excess calcium ions.

#### Cardiovascular Stem Cell Transplantation Therapies

MI commonly known as a heart attack, represents a significant cause of mortality and morbidity globally [190]. The insufficient blood flow and lack of oxygen reaching the heart muscle downstream during MI can trigger the programmed cell death (apoptosis) and premature cell death (necrosis) of cardiomyocytes (CMs). This process significantly contributes to the development of heart failure, affecting a considerable proportion of individuals post-MI, estimated at 10–40% [191].

Research efforts were devoted to counteracting the permanent cell damage caused by reduced blood flow in the heart during a heart attack or in the blood vessel lining during peripheral arterial disease (PAD). A relevant approach relies on the injection of stem cells (mainly MSCs [192]) directly into the affected area, aiming to potentially replace lost cells and stimulate the body’s inherent tissue repair mechanisms by releasing growth factors that promote regeneration [193].

An important application of injectable alginate hydrogels is in the treatment of MI [153,194,195,196,197,198,199,200,201]. The role of alginate as a supportive matrix for MSCs is crucial for immune protection and enhancing cell viability and retention in the cardiac microenvironment [196,197,202,203,204,205] which presents high oxidative stress and poor nutrient supply. Several reports pointed toward high cell death rates (90–95%) shortly after injection of cell suspensions in aqueous media [206,207]. Additionally, less than 3% of MSCs are retained in the infarcted heart 24 h after transplantation [208]. Injectable alginate hydrogel formulations of MSCs improved cell retention rates up to 50–62% [179,209], and promoted enhanced cell proliferation and contractile function [195,210,211]. Notably, crosslinked alginate hydrogel has mechanical properties closely similar to the cardiac tissue [212] which further emphasize its suitability for the treatment of MI. Interestingly, as the post-infarction environment undergoes recovery, the concentration of local calcium ions decreases. This reduction leads to an increased rate of alginate dissociation and clearance from the region [211].

In order to improve injectable alginate hydrogel formulations for the treatment of MI, various methodologies have been investigated. These include modifying alginate to improve its mechanical properties and electrical conductivity [213,214]. For instance, Liang et al. reported on a hydrogel produced from oxidized alginate chemically crosslinked with gelatin intended for the transplantation of adipose-derived stem cells (ADSCs) for treating MI in rats [215]. In this study, a multifunctional polymer based on tetraaniline (TA) and 2-aminopyridine-5-thiocarboxamide (APTC) was incorporated in the hydrogel formulation to ensure conductive properties, inhibit inflammatory responses and stimulates angiogenesis by a sustained H_2_S release after cell transplantation (Figure 5a–c). In fact, H_2_S demonstrated favorable effects by reducing the size of damaged heart in MI and improving fractional shortening (FS) [216]. It is noteworthy to mention that the mechanical properties of the alginate-gelatin hydrogels were adequate for stable anchoring to wet and beating hearts.

The modification of alginate with the cell-binding ligand RGD significantly improved hMSC retention. For instance, the injection of RGD-modified alginate resulted in a notable increase in hMSC retention, reaching 60%, as compared to 9% retention observed in saline controls [179].

Additionally, composites based on alginate hydrogels have been prepared to reduce oxidative stress damage to transplanted cells in the presence of reactive oxygen species (ROS) in the MI microenvironment [217,218].

For instance, Hao et al. developed alginate-based hydrogels integrating brown adipose-derived stem cells (BADSCs) and fullerenol, known for its antioxidant activity [217]. The incorporation of fullerenol nanoparticles in the hydrogel formulation contributed to the inhibition of oxidative stress damage, the activation of extracellular regulated protein kinase and p38 pathways in the ROS microenvironment, and the suppression of JNK signaling pathways. The involvement of BADSCs in the recovery of the infarcted heart was demonstrated through observable angiogenesis and restructuring of the left ventricle. Furthermore, the hydrogel provided long-term mechanical and physical support to damaged cardiac tissue and increased ventricular wall thickness to alleviate myocardial stress (Figure 5d–j).

Liu et al. recently developed an injectable hydrogel composed of alginate and fibrin to enhance the survival and integration of transplanted cardiomyocytes (CMs) for MI therapy [219]. The introduction of fibrin enhanced CM adhesion, thereby promoting their retention within the hydrogel matrix. Also, their study involved the incorporation of VEGF into the hydrogel, aiming to promote angiogenesis within the infarcted area and improve the ischemic conditions in the local microenvironment. Cardiac function was assessed via echocardiography at 7 and 28 days following hydrogel injection. The formulations containing transplanted cells and VEGF exhibited significant effectiveness in enhancing cardiac function, resulting in heightened blood vessel density, reduced infarct size, and improved overall cardiac performance.

#### Traumatic Brain Injury

TBI stands as one of the most prevalent forms of neurotrauma, bearing significant morbidity and mortality rates, with estimates reaching over 50 million cases annually worldwide [220,221]. Indeed, despite the absence of a specific drug or clinical treatment capable of reversing the pathological process or significantly improving the remaining dysfunctional nerve [222], recent advancements have highlighted MSCs transplantation as a promising therapeutic avenue for TBI treatment [223]. These transplanted cells display the capacity to differentiate into neural cells, replace damaged cells, and release paracrine factors, exhibiting substantial potential in the treatment of TBI.

Ma et al. developed an injectable hydrogel combining alginate and collagen, integrating BMSCs for the treatment of TBI [176]. This formulation enabled sustained release of stromal cell-derived factor-1 (SDF-1), creating a biocompatible and biodegradable microenvironment which supported the survival, migration, and neuronal differentiation of BMSCs in vitro.

The combination of hyaluronic acid and alginate holds promise for formulating injectable hydrogels. For instance, Zhang et al. explored loading human umbilical cord mesenchymal stem cells (hUC-MSCs) into alginate and hyaluronic acid blends with varying ratios [224]. An optimal ratio alginate:HA of 2:1 provided superior protection for hUC-MSCs, possibly due to the incorporation of HA, which facilitated enhanced cell attachment. In vivo evaluation in SD rats pointed toward the most pronounced motor ability recovery for specimens having received hUC-MSCs loaded hydrogels.

#### Bone Regeneration

In cases of large bone defects resulting from trauma, disease, or malformation, cell transplantation emerged as a promising therapeutic approach to facilitate bone healing. Alginate-based injectable hydrogels have consistently played a crucial role for the development of bone regeneration materials, allowing effective filling of irregular bone defects and reduced surgical intervention [225].

Ingavle et al. developed a peptide-modified composite hydrogels, combining alginate and hyaluronate modified with RGD, to encapsulate MSCs for bone repair purposes [226]. This newly formed RGD-alginate/RGD-hyaluronate (RGD-SA-HA) hydrogel demonstrated enhanced attraction and proliferation of ovarian MSCs in vitro compared to single component hydrogels. Additionally, hydrogels incorporating mineralized polymeric microspheres enhanced their osteoconductive properties, thereby promoting and facilitating bone healing processes as demonstrated in a sheep bone defect model.

Liu et al. developed an innovative nanocomposite injectable hydrogel using LAPONITE^®^ XLG nanosilicates, alginate, and gelatin [227]. These nanosilicates play a pivotal role in the gelation process by forming hydrogen bonds with polysaccharide matrices, serving as a rheology modifier for alginate solutions, improving the hydrogel mechanical strength, and stimulating the osteogenic differentiation of hMSCs in vitro. The bioinert properties of alginate contributed to the in vivo durability of the hydrogel, preventing premature degradation by proteases and ensuring a scaffold resorption rate aligning with new bone tissue formation. In their study, the hydrogel loaded with rat bone marrow mesenchymal stem cells (rBMSCs) significantly enhanced bone healing in rat calvarial defects compared to non-cellularized hydrogel. Importantly, this approach exhibited no adverse effects on the recipients, indicating its potential for augmenting bone regeneration.

Table 2 summarizes a variety of studies that involve encapsulating cells using injectable hydrogel.

### 3.2. Microencapsulation

Cell microencapsulation involves enclosing cells within a suitable material to create spheres in the micrometer range (microspheres). Unlike macrodevices, these microspheres are generally not retrieved once transplanted. This approach offers specific advantages over macroscopic scale devices. Their smaller size results in a higher surface-to-volume ratio, allowing more efficient exchange of nutrients and metabolites crucial for the viability and the therapeutic functionality of the encapsulated cells. Furthermore, the reduced cell density within microspheres minimizes cell competition for nutrients, thus decreasing the risk of cell necrosis [232]. The most prominent method used for the production of microspheres is the extrusion of an alginate solution through a syringe into a gelation bath containing divalent cations inducing gelation of the droplets through ionic crosslinking. The mild gelation conditions of alginate microspheres led to the extensive use of this material for cell microencapsulation. However, translation into clinical practice is still facing a number of unmet challenges, including the optimization of the mechanical properties, the modification of the matrix activity and permeability, and the control over capsule size and size distribution uniformity. The extensive developments in both materials and cell encapsulation techniques using alginate-based matrices have been covered in recent reviews and are summarized below.

The use of alginate microspheres as delivery vehicles for cell-based therapies in the context of tissue engineering and regenerative medicine was broadly discussed by Xu et al. (2021) [233]. This review comprehensively covers several fabrication methods for alginate microspheres and their applications in cell delivery for the treatment of various diseases such as diabetes, bone and cartilage regeneration and vascularization. Recent advances in enhancing efficacy and biosafety of cell therapy through microencapsulation are covered by Lopez-Mendez et al. (2021) [234], who highlighted the development of materials and methods to improve biocompatibility, the importance of selecting suitable cell sources, targeting specific pathologies and the challenges of achieving optimal microspheres size and shape. Likewise, Ashimova et al. (2019) [235] focused on the immunological challenges associated with cell microencapsulation, reporting approaches to mitigate immune responses using highly purified materials and incorporating immune modulators. A wider view of alginate microspheres is discussed by Dhamecha et al. (2019) who summarized the main applications of alginate-based microspheres in therapeutic delivery and cell culture [236]. Focusing on the development of new method for the engineering of microspheres, Mohajeri et al. (2022) [237] discussed the use of droplet-based microfluidics in the context of cell microencapsulation within alginate-based matrices. Compared to extrusion technologies, droplet-based microfluidics result in more controlled and precise droplet formation, leading to more consistent microspheres size and structure, thereby enhancing their potential for cell therapy applications. Chemical alterations of alginate for cell microencapsulation are treated in the review of Len’shina et al. (2021) [238]. These molecular modifications aim to reduce immune response and fibrosis, ensuring longer stability and survival of encapsulated cells. The challenges and advancement made toward the use of microencapsulation for cell therapies in specifically managing type 1 diabetes are discussed in two recent reviews by Basta et al. (2021) [239] and Zhang et al. (2022) [112], with emphasis on biocompatibility, immune protection, and efficiency of encapsulated islet grafting procedures. Finally, Lopez-Mendez et al. (2021) [240] gathered the progress made in clinical trials and companies involved in the field of cell microencapsulation for the treatment of type 1 diabetes, cancer and CNS disorders.

These reviews collectively underscore the challenges and advances made toward ensuring successful cell microencapsulation within alginate matrices. Still, there remains a need for tailored microspheres in specifics applications. The following section introduces some of the latest, highly specific studies in that domain.

Improving the in vivo durability of microspheres is crucial for the efficacy of cell therapies requiring long-term residence and functionality of the transplanted cells. The combination of alginate with other polymers, either through layering or intertwining of the components, is a common strategy to maintain the integrity of the resulting microspheres in physiological environment. Following this approach, alginate/poly-L-ornithin polyelectrolyte hydrogel were designed to form microspheres through aqueous electrospraying technique, which allowed precise control over the hydrogel properties [241]. Using the same approach, gelatin methacryloyl-alginate core–shell microspheres were investigated for endodontic regeneration. These microspheres, produced via coaxial electrostatic microdroplet method, contained human dental pulp stem cells (hDPSC) and human umbilical vein endothelial cells (HUVEC). It demonstrated enhanced proliferation and vasculogenic and osteo/odontogenic potentials in co-culture conditions, leading to the formation of prevascularized microtissues [242]. Also, to improve the stability and longevity of microspheres in the context of type 1 diabetes treatment, three layers microspheres composed of alginate-poly-L-ornithine-alginate were designed using methacrylated alginate covalently conjugated at the outer layer. These systems exhibited increased stability in vitro and in vivo but were not evaluated with immobilized cells [243]. However, as recently discussed by Distler et al. (2021) [244], the presence of microencapsulated cells can greatly influence the mechanical properties of the hydrogel. They demonstrated the effect of murine embryo fibroblast loading on the mechanical behavior of oxidized alginate-gelatin hydrogels. At high cell concentrations, the hydrogel stiffness decreased and the material exhibited more pronounced nonlinearity under larger strains and faster stress relaxation. Therefore, considering cell concentration within the microdevices is crucial for determining the final mechanical properties of hydrogels.

Continuous efforts were also produced to enhance the biocompatibility of alginate-based microspheres. The incorporation of sulfated alginate in the composition of multilayer capsules resulted in fibrosis mitigation [245] The co-encapsulation of bioactive compounds with cells was also explored, pointing toward the beneficial effect of curcumin on inflammation and fibrosis following the implantation of alginate encapsulated pancreatic beta cells [246].

While controlling the host response to implanted materials is crucial, the compatibility of the hydrogel matrix toward the encapsulated cells is another key parameter to ensure graft survival and functionality. Crisóstomo et al. demonstrated that controlling the alginate matrix stiffness through the adjustment of divalent cations concentration, in combination with incorporation of pancreatic extracellular matrix, improved the insulin secretion of microencapsulated islets in response to glucose stimulation [247]. Similarly, co-encapsulation of beta cells and nanoparticles containing glucagon-like peptide-1 (GLP-1) in alginate hydrogels increased the metabolic activity and insulin secretion of the encapsulated beta cells [248]. These recent studies highlight promising strategies to enhance the functionality, effectiveness and longevity of cell-based therapies for the treatment of type 1 diabetes.

Lastly, the variety of microencapsulation methods continues to expand as specific applications require specialized techniques. In this respect, a recent study investigated a new approach for encapsulating human bone marrow-derived mesenchymal stem cells (hBM-MSCs) in alginate microspheres. This method utilizes one-step emulsification by internal gelation to create small-sized microspheres (127–257 µm), which offer improved mass transfer and mechanical strength while maintaining good cell viability and functionality in vitro and in vivo [249].

### 3.3. Nanocrocapsulation (Layer-by-Layer (LbL) Self-Assembly)

LbL self-assembly is a versatile technique for the encapsulation of living cells. It involves the deposition of two oppositely charged polyelectrolytes onto the cell surface [250,251,252,253]. This method finds applications in diverse areas such as cell protection, cell transplantation, cell-based biosensors and the regulation of cell proliferation and differentiation [254].

This process results in the creation of an extracellular coating which protects the cells from physical damage and provides a microenvironment favorable to cell integration and sustained functionality. Importantly, this encapsulation approach preserves cell viability and proliferation without adverse effects such as inflammatory reactions [255,256]. Representative compositions involve the coating of negatively charged islets [257] with positively charged polymers, such as cationic gelatin [256,258,259] and chitosan [260], followed by subsequent coating with anionic polymers (e.g., alginate, heparin, hyaluronic acid) [256,258,259,261]. This approach minimizes the required transplant volume for cell transplantation by employing nanometer-sized ultrathin layers (<50 nm) of polymer coating [250]. In addition, the nanoscale isolation layer decreases the diffusion distance between the encapsulated islets and the host environment, thereby enhancing the efficacy of the transplantation process [262]. The thickness of the polymer shell [261] and the incorporation of diverse functionalities, including drug cargos, through multiple layers [262,263] can be easily controlled in LbL self-assembly processes.

Thus far, most applications of LbL self-assembly focused on the encapsulation of pancreatic islets [253,263,264,265,266,267,268,269] and neural stem cells [259,261]. Combination of countercharged gelatin and alginate stands as the most popular composition for the encapsulation of mammalian cells as these polymers display opposite charges under the same pH conditions [270], as well as high cell compatibility. For example, Li et al. demonstrated that LbL encapsulation using gelatin and alginate polyelectrolytes (average thickness of 6 ± 2.3 nm) did not compromise the viability, proliferation, or differentiation capabilities of NSCs [259]. This study highlighted the beneficial effect of loading IGF-1 within the polymeric matrix to enhance the properties of NSCs due to sustained release under acidic conditions. This suggests potential applications in specific nervous system disorders.

In a different approach using LbL technology, fibroblast growth factor-2 (FGF-2) was incorporated into the gelatin component of the multilayered coating to support the survival and proliferation of dermal papilla cells [271].

The LbL gelatin-alginate encapsulation was applied by Ru et al. to the transplantation of retinal pigment epithelial (RPE) cells into the subretinal space of Royal College of Surgeons (RCS; retinal dystrophic) rats, without the use of immunosuppressive medication [272]. While previous studies indicated the low engraftment capacity [273] and uncontrolled distribution of injected cell suspensions [274], this technique showed successful engraftment of RPE cells, minimized diffusion and reduced immunogenicity.

However, limitations associated with the LbL encapsulation approach include the relatively loose structure of the encapsulation shell, which can lead to premature dissolution of the polymeric protective layer. This technique is therefore not recommended for applications requiring long-term integrity of the protective shell. Additionally, there is a concern regarding the cytotoxicity of the polyelectrolytes employed in the process [275]. Recent studies suggested that the integration of thiol, acrylate, hydroxyl and aldehyde groups on the alginate backbone stabilized the encapsulating multilayered coating while preserving the material permeability [276,277].

A compilation of various studies devoted to cell encapsulation using LbL self-assembly is provided in Table 3.

## 4. Conclusions

Cell transplantation stands as a promising medical procedure aimed at replacing damaged or dysfunctional cells, restoring function, and promoting tissue repair. A notable example of its potential lies in the treatment of degenerative diseases like type 1 diabetes, where promising results from clinical studies have been observed. The field of cell transplantation has witnessed significant progress, owing much to the use of biomaterials that protect cells and provide a favorable environment for their survival, growth, and functionality after transplantation. Alginate, with its versatile, biocompatible, and tunable properties, has emerged as a key biomaterial driving progress in cell transplantation.

A considerable number of studies have examined the use of alginate microspheres in diverse cell transplantation applications, with the majority of these reports concentrating on the treatment of type 1 diabetes. The significance of alginate microspheres lies in providing a high surface area for improved exchange of oxygen and nutrients toward immobilized cells. However, one limitation of this approach for cell transplantation could be the inability to retrieve the microspheres. On the other hand, there has been notable advancement in the design of macroencapsulation devices that not only facilitate retrieval after transplantation but also contribute to vascular promotion, in contrast to microspheres. Certain designs for macroencapsulation utilize fibers with excellent mechanical properties, such as nylon, which is coated with a thin film of alginate containing targeted cells [65,90,100]. Additionally, other devices are based on nanofibers that form a permeable membrane, subsequently coated with an alginate solution mixed with cells [70,71,72,73]. Some retrieval devices contribute significantly in establishing vascularization in host tissue, which is a critical aspect for the success of numerous cell transplantation therapies.

Designing anisotropic capillary structures based on alginate has emerged as a promising approach for directing the growth of axons, particularly in the context of treating SCI [74].

The use of 3D printing technology for designing cell encapsulation devices based on alginate bioinks offers a number of advantages, including precise control over the spatial distribution of cells and biomaterials and the possibility to engineer complex, three-dimensional structures that mimic native tissues. For instance, one advanced 3D-printed construct involved the preparation of an alginate core–shell structure loaded with multiple cell types [107]. Alternatively, the LbL deposition of bioinks with different cell type contents resulted in high precision structures [136]. In these approaches, channels or networks can be incorporated into the printed structures, facilitating the formation of functional blood vessels within the tissue constructs [120]. The challenge associated with the exchange of oxygen and nutrients for cells due to low surface area could be a limitation for such devices. Ongoing research in this field appears to focus on using various bioink formulations with different cell types to support optimal cell survival. Additionally, there is a focus on designing innovative architectures of macroencapsulation systems to enhance the exchange of oxygen [150].

Injectable alginate-based hydrogels are another promising method in advancing the field of cell transplantation. The majority of reported studies focused on treating conditions such as MI, SCI, and TBI. The primary rationale behind using injectable alginate hydrogels lies in their ability to conform to irregular shapes, ensuring optimal contact between the hydrogel and the damaged tissue, for example, myocardium or contusive lesion in cases of MI and SCI, respectively. Additionally, employing injectable hydrogels can minimize the invasiveness of cell delivery to targeted damaged tissues.

Therefore, the utilization of alginate in various forms, such as solid scaffold, microspheres, fiber, bioinks and injectable hydrogels, provides a highly promising biopolymer for cell transplantation.

## Data Availability

No new data were created in this study. Data sharing is not applicable to this article.

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
