# Peer review of "Recent Advances in Alginate-Based Hydrogels for Cell Transplantation Applications"

_pharmaceutics, 2024, doi:10.3390/pharmaceutics16040469_

Round 1

Reviewer 1 Report

Comments and Suggestions for Authors

The paper presents recent results regarding the use of alginate for cell entrapment/encapsulation for their transplantation.

 The chapter numbering should be revised. See also chapter “3. Nanocrocapsulation”.

The term “cell encapsulation” is usually used for the formation of capsules, meaning the immobilization of cells within a polymeric semi-permeable membrane. The entrapment of cells in different matrices can be considered encapsulation?

Microneedles are presented in Table 1 and Graphical abstract, but not in the text.

Lines 532-533: Layer by layer-by-layer technique can be confused with layer-by layer self-assembling technique. Reference 142 said that “Bioprinting technology is an innovative process in advanced tissue engineering that allows the layer-by-layer arrangement of a wide array of cell types”.

The formulations for the injectable hydrogels presented in Table 2 should be checked. In ref. 202, the alginate was modified with a peptide sequence, and CaSO4 was used to crosslink the alginate. In ref. 170, injectable hydrogel also contains FeCl3 for the ionic crosslinking of alginate. The ions used for the formation of the injectable hydrogels can be also presented.

Comments on the Quality of English Language

-

Author Response

Please see enclosed document.

Reviewer 2 Report

Comments and Suggestions for Authors

I have gone through the review entitled "Recent advances in alginate-based hydrogels for cell transplantation applications".

This review seems interesting in terms of the uses of alginate materials, and their formulations, in regenerative medicine. Particularly, the authors describe different techniques for cell encapsulation, focusing on the use of alginate as an exceptional biomaterial for cell transplantation.

The level of details, figures and chemical information is well explained/presented for different alginate materials.

Some minor points need to be corrected:

-Table 3, check the font size for the footer of the table (PLL: poly(l-lysine)).

-There is an additional space on lines 996-997.

-Considering the relevance of nanogels in materials science, are there reports of alginate-based nanogels for cell encapsulation? it would be good to do some searching and incorporate it into the review.

-For this type of materials, are there recommended studies to evaluate their stability and/or durability and their metabolization for in vivo studies?

Author Response

Please, see enclosed document.

Reviewer 3 Report

Comments and Suggestions for Authors

Manuscript ID: pharmaceutics-2915907

Title: Recent advances in alginate-based hydrogels for cell transplantation applications.

Authors: Alireza Kavand, François Noverraz, Sandrine Gerber-Lemaire.

 In the paper titled "Recent Advances in Alginate-Based Hydrogels for Cell Transplantation Applications," the authors conducted a comprehensive review that summarizes the diverse applications of alginate, a highly biocompatible biomaterial, in the field of regenerative medicine, specifically in cell transplantation applications. They provide an in-depth exploration of alginate's use in various formulations, including microparticles, injectable hydrogels, rigid scaffolds, and bioinks, for cell encapsulation and transplantation. The review offers valuable insights into the efficacy of alginate-based therapies for treating a wide range of diseases and injuries in regenerative medicine.

 While the review provides comprehensive information, the structure could benefit from clarification as some subsections are unclear. Additionally, attention to wording and punctuation would improve readability. Enhancing the conclusions section with further insights would strengthen the manuscript's impact. Overall, I think the review contributes significantly to the field and can be considered suitable for publication in Pharmaceutics after minor revisions. The authors should respond or correct the manuscript in a positive way to the following issues:

 Structure

        1. The numbering of sections seems to be incorrect, leading to confusion for readers. In its current form:

 1.  Introduction

Alginate

Cell encapsulation techniques

1.  Macroencapsulation

1.1.   Macroscopic devices

1.1.1. Device based on porous membranes

1.1.2. Anisotropic 3D scaffolds

1.1.3. Hydrogel Fibers

1.1.4. 3D printing devices

1.1.5. Injectable hydrogels

      Nervous System Stem Cell Transplantation Therapies

      Cardiovascular Stem Cell Transplantation Therapies

      Traumatic brain injury

      Bone regeneration

2.  Microencapsulation

3.  Nanocrocapsulation (Layer-by-layer (LbL) self-assembly)

5.  (!) Conclusions

        Authors are encouraged to review and correct the section numbers and/or consider adjusting the overall section format for clarity and consistency.

        2. In line with the previous comment, the manuscript appears to disproportionately emphasize macroencapsulation over micro and nano approaches. Balancing the focus across these different scales of encapsulation would provide a more comprehensive overview of the field.

        3. Line 515 - While it's acknowledged that low-concentration GelMA hydrogels may pose limitations, it's debatable whether the claim “low concentration GelMA hydrogels are not suitable for the regular printing of cell-laden fibers” (line 515). The following paper https://doi.org/10.1002/adfm.202208940 is a recent and good example. It showcases a strategy for encapsulating cells in GelMA for bioprinting, contradicting the notion that low-concentration GelMA hydrogels are universally unsuitable for this purpose, as it may depend on specific bioprinting techniques employed.

        4. It would be beneficial for the authors to reconsider the formulation of the Conclusions section. It should not only provide a summary but also interpret findings, discuss significance, address limitations, and suggest future directions. This will enhance the manuscript's impact and value for readers.

 Language and vocabulary

        5. In terms of vocabulary and language, the authors are encouraged to revisit the wording throughout the document, particularly in instances where phrases like "crucial role" and similar expressions appear repetitively. Diversifying the language and incorporating more nuanced descriptors would enhance the clarity and richness of the manuscript's narrative.

Formatting and style

        6. There is inconsistency in the use of bold font when referring to figures.

        7. Similarly, there is a lack of uniformity in the placement of periods before or after reference numbers throughout the document.

        8. Figure 1 lacks scales for reference, which are essential for accurately interpreting the depicted data. Including numerical values or units, alongside the axes of the graph would provide readers with a clear understanding of the measurements or quantities represented in the figure.

        9. The captions accompanying Figure 5 may benefit from adjustments in size as they appear challenging to read.

        10. The manuscript contains several typographical errors; some examples are found in lines:

 -   Line 332: facilitating easy portability...

-   Line 486: collagen, agarose, , gelatin...

-   Line 730: alginate and fibrin as two natural biomaterials to enhance the survival and integration of transplanted.

             It is recommended to carefully examine the manuscript to make the necessary corrections.

Comments on the Quality of English Language

Minor editing of English language required.

Author Response

Please, see enclosed document.
